# Ringer’s Lactate Prevents Early Organ Failure by Providing Extracellular Calcium

**DOI:** 10.3390/jcm9010263

**Published:** 2020-01-18

**Authors:** Biswajit Khatua, Jordan R. Yaron, Bara El-Kurdi, Sergiy Kostenko, Georgios I. Papachristou, Vijay P. Singh

**Affiliations:** 1Department of Medicine, Mayo Clinic, Scottsdale, AZ 85259, USA; 2Department of Medicine, Ohio State University College of Medicine, Columbus, OH 43210, USA

**Keywords:** pancreatitis, organ failure, Ringer’s lactate, inflammation, CRP, calcium, lipolysis, saponification, isothermal titration calorimetry, mitochondrial depolarization

## Abstract

Objective: Ringer’s lactate may improve early systemic inflammation during critical illnesses like severe acute pancreatitis, which are associated with hypocalcemia. Ringer’s lactate is buffered and contains lactate and calcium. We, thus analyzed extracellular calcium or lactate’s effects on the mechanisms, intermediary markers, and organ failure in models mimicking human disease with nonesterified fatty acid (NEFA) elevation. Methods: Meta-analyses and experimental studies were performed. Experimentally, extracellular calcium and lactate were compared in their interaction with linoleic acid (LA; a NEFA increased in human severe pancreatitis), and its subsequent effects on mitochondrial depolarization and cytosolic calcium signaling resulting in cell injury. In vivo, the effect of LA was studied on organ failure, along with the effect of calcium or lactate (pH 7.4) on severe acute pancreatitis-associated organ failure. A meta-analysis of human randomized control trials comparing Ringer’s lactate to normal saline was done, focusing on necrosis and organ failure. Results: Calcium reacted ionically with LA and reduced lipotoxic necrosis. In vivo, LA induced organ failure and hypocalcemia. During severe pancreatitis, calcium supplementation in saline pH 7.4, unlike lactate, prevented hypocalcemia, increased NEFA saponification, reduced circulating NEFA and C-reactive protein, reduced pancreatic necrosis adjacent to fat necrosis, and normalized shock (carotid pulse distension) and blood urea nitrogen elevation on day 1. This, however, did not prevent the later increase in serum NEFA which caused delayed organ failure. Meta-analysis showed Ringer’s lactate reduced necrosis, but not organ failure, compared with normal saline. Conclusion: Hypocalcemia occurs due to excess NEFA binding calcium during a critical illness. Ringer’s lactate’s early benefits in systemic inflammation are by the calcium it provides reacting ionically with NEFA. This, however, does not prevent later organ failure from sustained NEFA generation. Future studies comparing calcium supplemented saline resuscitation to Ringer’s lactate may provide insights to this pathophysiology.

## 1. Introduction

Critical illnesses such as severe acute pancreatitis can cause organ failure [1,2]. These are commonly associated with hypocalcemia [3,4], which is also a part of Ranson’s [5] and Glasgow criteria [6] for predicting severe pancreatitis. Ringer’s lactate (RL) is buffered and contains calcium and lactate. Clinical studies show RL infusion during pancreatitis allows early improvement in systemic inflammatory response syndrome (SIRS) and C-reactive protein (CRP) when compared with saline infusion [7,8,9], but not survival [10]. RL’s benefits have been mechanistically attributed to lactate inhibiting macrophage activation [7] or negatively regulating toll-like receptor 4(TLR4) in a G protein-coupled receptor 81(GPR81)-dependent manner [11]. The cell and animal models used previously, however, do not clinically mimic a severe disease with associated organ failure.

Interestingly, previous studies have shown that RL infusion in humans does not influence serum lactate concentrations alone or in comparison to saline [12,13], bringing into question a direct role of lactate in mediating the benefits of RL. Lactate infusion reduces the drop in serum bicarbonate [8] and may protect from acidosis. Recently, a Cochran database study of evidence concluded that buffered solutions do not provide any benefit over 0.9% saline in reducing in hospital mortality among critically ill patients with a high degree of certainty [14]. Collectively, these suggest that the benefits of RL are noted in early severe pancreatitis and may be from a mechanism other than its being buffered or containing lactate.

Fatty acids are elevated during critical illnesses [15,16,17,18,19] such as severe acute pancreatitis in humans [20,21] and rodents [22,23,24,25], but not during mild disease [22,23]. Previous studies have shown hypocalcemia to result from fatty acid administration [26] and calcium to be present in the (peri) pancreatic fat necrosis associated with pancreatitis [24], consistent with saponification [24,27,28]. It is therefore plausible that the benefits noted with RL could be because of the 3 mEq. calcium it contains. We thus investigated whether the extracellular calcium supplementation is beneficial during severe pancreatitis associated with organ failure, mimicking severe human disease, and compared its efficacy to lactate. We also compared extracellular calcium’s benefits in neutralizing fatty acid signaling and to the pathophysiology of hypocalcemia. These findings were analyzed in the context of a new meta-analysis of the three randomized trials comparing RL to normal saline which focused on organ failure, ICU requirements, and also to a previous meta-analysis [10] that focused on CRP and SIRS and mortality. Interestingly, we note that the extracellular calcium (CaE) has a mechanistically distinct role, independent of lactate or pH, in reducing lipotoxic necrosis and transiently improving SIRS in severe pancreatitis, similar to what is observed in the clinical studies.

## 2. Methods

All studies were approved by the Institutional Animal Care and Use Committee at the Mayo Clinic.

### 2.1. In Vitro Studies

Isothermal Titration Calorimetry (ITC): The interaction of calcium or lactate with linoleic acid was carried out on a Nano ITC (TA Instruments-Waters LLC, New Castle, DE, USA) instrument using Nano ITCRun Software v3.5.6. The experimental setup was equipped with a computer-controlled micro-syringe injection device. A total of 25 aliquots of freshly prepared, degassed calcium chloride or sodium lactate (1 mM in 10 mM HEPES [4-(2-hydroxyethyl)-1-piperazineethanesulfonic acid], pH 7.4; 1.7 µL per injection) were injected from a rotating syringe of speed 350 rpm into the ITC sample chamber containing freshly prepared, degassed Linoleic acid (1 mM in 10 mM HEPES, pH 7.4) equilibrated at 37 °C. The interval between two injections was 200 s. Similarly, calcium chloride injection into HEPES buffer, HEPES buffer into linoleic acid, and HEPES into HEPES were run as controls. A representative thermogram from three independent experiments is shown in Figure 1.

The thermodynamic parameters (Kd, *n*, ΔH, and ΔS) for calcium–linoleic acid, calcium-lactic acid, or lactate–linoleic acid were calculated from NanoAnalyze Software v3.10.0 (TA Instruments-Waters LLC, New Castle, DE, USA). Parameters were represented as Mean ± SD (standard deviation).

### 2.2. Primary Acinar Cell Preparation

16 week old male CD-1 mice were euthanized by CO_2_ asphyxiation with secondary euthanasia by cervical dislocation. Whole pancreata were removed, trimmed of excess fat, and placed in oxygenated HEPES buffer [29] (10 mmol/L HEPES at pH 7.4, 130 mmol/L NaCl, 5 mmol/L KCl, 1 mmol/L MgCl_2_, 1 mmol/L CaCl_2_, 10 mmol/L glucose, 10 mmol/L sodium pyruvate, and 0.1% bovine serum albumin) on ice. Pancreata were insufflated with collagenase solution, minced, and incubated in collagenase at 37 °C with shaking (120 RPM) for 40 min [29]. Digested tissue was triturated with a 10 mL serological pipette, filtered sequentially through a 180 micron steel mesh and a 70 micron cell strainer to obtain acinar cells. Cells were washed twice by low-speed centrifugation with fresh, oxygenated HEPES 0.01% albumin without collagenase, and suspended in this buffer for use. Cell viability was verified >95% by trypan blue exclusion prior to experimentation.

### 2.3. Lactate dehydrogenase (LDH) Release Assays

Primary acinar cells were seeded in a 12-well plate using a wide-bore 1 mL micropipette tip in HEPES buffer with the indicated concentrations of calcium, BAPTA-AM [1,2-Bis(2-aminophenoxy)ethane-N,N,N′,N′-tetraacetic acid tetrakis(acetoxymethyl ester); 50 μM], dantrolene (100 μM), thapsigargin (1 μM), or EGTA (ethylene glycol-bis(β-aminoethyl ether)-N,N,N′,N′-tetraacetic acid; 1 mM), or were seeded in Ringer’s lactate buffer pH 7.4 as described in the results. Ringer’s lactate has pH ≈ 6.8 (6.0–7.5), which was adjusted to 7.4 by adding NaOH (1 N) using a pH meter before use. Cells were treated with 100 μM final concentration linoleic acid (LA) sonicated into HEPES buffer or 100 nM final concentration caerulein and incubated at 37 °C with shaking at 40 RPM. Indicated time-point supernatants and 1% Triton X-100 total cell lysates (taken as 100%) were collected for LDH analysis with the LDH Cytotoxicity Assay PLUS kit (Roche, Basel, Switzerland) according to manufacturer’s protocol and read on a FlexStation 3 Multi-Mode Microplate Reader (Molecular Devices, San Jose, CA, USA).

### 2.4. Fura-2/JC-1 Spectrophotometry

Fura-2 AM and JC-1 at 5 μg/mL final concentration in 0.5% dimethyl sulfoxide were sonicated into HEPES buffer to obtain a working staining solution. Live acinar cell preparations were washed and resuspended in staining solution and incubated at 37 °C with slow rotation protected from light. Cells were gravity-settled and washed twice with fresh HEPES buffer. Stained cells were suspended in a UV/VIS-transparent cuvette containing a stir bar and placed in a F2100 Hitachi Fluorescence Spectrophotometer (Hitachi, Tokyo, Japan). Fura-2 ratio (340/380 nm excitation, 510 nm emission) and JC-1 ratio (510 nm excitation, 530/590 nm emission) were collected every 15 s with stirring and temperature control as previously described [25].

### 2.5. In Vivo Experimental Models and Parameters

#### 2.5.1. Linoleic Acid Administration

Linoleic acid (LA) was administered intraperitoneally at 0.1% body weight to adult CD-1 mice (30–40 g) as described previously [25]. These were monitored for vitals daily before and after LA administration as described below. These were followed till 72 h after, or until moribund, whichever came first, at which time they were euthanized as described below.

#### 2.5.2. Acute Pancreatitis Model

11–12 week old, 54 ± 2 g, *ob/ob* [B6.Cg-*Lep^o^^b^*/J; Jax Stock #00632, Jackson Laboratory (Bar Harbor, ME, USA)] mice were used for Caerulein (CER)-induced acute pancreatitis study, as described previously [23]. We had four groups (8–10 animals per group). These were control, CER, CER + calcium chloride, and CER + sodium lactate. A baseline blood sample was first collected from the tail vein, and vitals were measured. Pancreatitis was then induced by planting a primed Micro-Osmotic pump (Alzet, DURECT Corporation, Cupertino, CA, USA) to deliver CER (50 mcg/kg/h; Bachem AG, Bubendorf, Switzerland) over 3 days. After 6–8 h of Alzet placement, a tail vain blood sample was collected to confirm >3-fold elevation of lipase over baseline. After that, calcium chloride (10 mM, 1 mL/dose, 3 times/day) or sodium lactate (30 mM, 1 mL/dose, 3 times/day) in saline, pH 7.4, were given intraperitoneally for 3 days (55 mL/kg/day) to the respective group of mice. The saline pH was ≈ 6.2 at baseline, and was adjusted to 7.4 using NaOH (1 N) and a pH meter. Animals were followed for up to 5 days or until they were moribund, whichever came first, at which time they were euthanized in a carbon dioxide chamber before harvesting blood and tissues.

#### 2.5.3. Monitoring

Effects of acute pancreatitis (AP)r LA administration on carotid pulse distention (measured in micrometers) were assessed with a collar sensor using the MouseOx oximeter (Starr Life Science, Pittsburgh, PA, USA) before induction of AP and daily thereafter. Mice were followed for up to 5 days for survival or were sacrificed earlier if fulfilling criteria for euthanasia (e.g., if moribund). Blood, pancreas, kidney, and lung tissues were harvested to study various parameters of pancreatitis.

### 2.6. Histology, Special Stains, and Immunohistochemical Studies

Pancreas, kidney, and lung tissues were fixed with 10% neutral buffered formalin (Fisher Scientific) and embedded in paraffin and sectioned. Whole pancreas paraffin section (5 microns) slides stained by hematoxylin and eosin (H&E) were used to quantify pancreas necrosis and peri-fat acinar necrosis (PFAN), that is, necrosed acinar area adjacent to fat necrosis as described previously [24,27], and slides stained by Von Kossa using a kit (Fisher Scientific, Hampton, NH, USA) to quantify fat necrosis and saponification as per recommended protocol.

### 2.7. Biochemical Assays

The biochemical assays (amylase, lipase, BUN [blood urea nitrogen], Calcium; Pointe Scientific, Canton, MI, USA) and colorimetric total free fatty acids using Non-Esterified Fatty Acid kit (FUJIFILM Wako Diagnostics, Mountain View, CA, USA) were done as per the manufacturer protocol and described previously [22,23,24,25,30]. Serum CRP level was measured by using Mouse C-Reactive Protein/CRP Quantikine ELISA Kit (R&D Systems, Minneapolis, MN, USA) as per manufacturer’s protocol.

### 2.8. Cytokine/Chemokine Assays

Serum Interleukin-6 (IL-6) and tumor necrosis factor-α (TNF-α) levels were assayed with a MILLIPLEX MAP Mouse Metabolic Magnetic Bead Panel (Millipore, Burlington, MA, USA) according to manufacturer’s recommendations on a Luminex 200 System (Life Technologies, Carlsbad, CA, USA) and analyzed using the xPONENT software as described previously [22,23,24,25,30].

### 2.9. Experiment Statistics

In vitro study data is from three or more separately done experiments, and in vivo data from 8–10 mice per experiment. Availability of tissue or sample (e.g., post mortem loss, volume remaining after other assays, space in a multi-well plate) dictated the number finally included in the analysis, with a goal to have at least seven randomly selected samples. Independent variables for in vivo and in vitro studies are shown as bar graphs or box plots, respectively, reported as mean ± SD. Line graphs ± SD were used for continuous variables. Significance levels were evaluated at *p* < 0.05. Data for multiple groups were compared by one-way ANOVA versus controls for that time point, and values significantly different from controls were shown as (*) unless otherwise mentioned specifically in the legend. When comparing two groups, a *t*-test or Mann–Whitney test was used, depending on the normality of distribution. Graphing was done using GraphPad Prism (version 8.0.0 for Windows, GraphPad Software, San Diego, CA, USA, www.graphpad.com).

### 2.10. Meta-Analysis

We conducted a comprehensive literature search and found that Iqbal et al. [10]. had recently published a systematic review and meta-analysis on the subject. We reviewed their data and found that while mortality was assessed as an outcome, no analysis was conducted to assess other outcomes such as pancreatic necrosis, intensive care unit (ICU) admission, and organ failure. We therefore extracted these data from the included papers and calculated overall odds ratio (OR) based on events/total ratios using the random effects model. Only randomized controlled trials were included in our analysis. Publication bias was assessed for using the Egger test. Statistical analysis was performed using the Comprehensive Meta-Analysis (CMA) software version 3.3.070 (Biostat; Englewood, NJ, USA). *p*-value <0.05 was considered significant.

## 3. Results

### 3.1. Calcium Binds Long-Chain Fatty Acids in an Energetically Favorable Manner

We first compared the energetics of fatty acid binding to lactate or calcium by isothermal titration calorimetry (ITC) in 10 mM HEPES (pH 7.4, at 37 °C). For this, each injection introduced 10 micromoles of calcium chloride (from a 1 mM stock in the burette) into 1 mM linoleic acid (LA). Linoleic acid was chosen since it is representative of the unsaturated fatty acids enriched in fat necrosis [22,24] and the sera of patients with severe pancreatitis [20], which, unlike saturated fatty acids, induce necrotic cell death [22,24,31]. The reaction was done in an adiabatically sealed calorimeter, and the heat transfer associated with the reaction was measured. As shown in Figure 1A, the interaction between calcium and LA was exothermic with large deflections initially. The enthalpy change (ΔH) of this interaction was about −20 KJ/mol (Figure 1B table, green line) and had a stoichiometry of about one Ca^2+^ for eight molecules of LA. In contrast, the injection of 10 micromoles sodium lactate into LA (redline, Figure 1B,C) had no increase in heat transfer over controls (black thermograms in Figure 1C). Similarly, injection of calcium into lactate was not different from to controls (data not shown). Thus, unlike lactate-LA, there was an electrostatic interaction between Ca and LA. We went on to study the biological relevance of this interaction.

### 3.2. NEFA Like Linoleic Acid Cause Hypocalcemia and Organ Failure

We next tested whether hypocalcemia may be relevant to the pathogenesis of NEFA-induced organ failure. Both oleic acid [32,33,34] and linoleic acid [25,30] can cause organ failure, as may happen in severe pancreatitis [22,25]. We thus analyzed the relevant parameters after administration of linoleic acid (0.1% body weight) in a cohort of mice we have previously studied [25]. The carotid pulse distention (pulse Dist.; Figure 1E; a large drop in which signifies shock) and rectal temperature (Figure 1F) were measured at 40 h, and the serum parameters were measured at the time of necropsy (46 ± 3 h). As can be seen, linoleic acid (LA) caused hypocalcemia (Figure 1D), associated with shock (Figure 1E), hypothermia (Figure 1F), and blood urea nitrogen elevation (BUN Figure 1G) consistent with multi-system organ failure. Thus, the Ca–NEFA interaction we note on ITC is biologically relevant, and results in hypocalcemia via binding LA, along with causing organ failure.

We next studied the mechanistic role of this interaction in pancreatitis by comparing extracellular calcium to lactate in reducing cell injury induced by a pancreatitis initiator (i.e., caerulein) versus that due to LA, which worsens pancreatitis via organ failure. The studies were first done in vitro and then in vivo. Major part of the in vitro studies was done in pancreatic acini, since these are susceptible to lipotoxic cell death [24,25] and acini also have receptors for caerulein, resulting in cell injury with high caerulein (≥1nM) doses.

### 3.3. Physiologically Relevant Extracellular Calcium (CaE), Unlike Lactate, Reduces NEFA Toxicity

Since LA is a NEFA increased in the sera [20,21] and pancreatic necrosis collections [22,24,35] during severe pancreatitis, burns [19] and critical illness [15], we studied the effect of extracellular calcium (CaE) on LA-induced cell injury [24,31]. We first tested the time dependence of incubating CaE with LA on LA-induced LDH leakage. Preincubating a 20× stock of LA (2 mM) and calcium (20 mM) before addition to acinar cells (final LA; 100 μM and calcium being 1 mM) time-dependently reduced LA-induced injury (Figure 2A). We then tested different pathologically and therapeutically relevant concentrations of CaE on LA-induced injury. A CaE of 2 mM significantly reduced LA-induced LDH leakage vs. lower CaE concentrations (Figure 2B). Substitution of HEPES buffer for Ringer’s lactate did not change the net LDH leakage induced by LA (47 ± 6%) compared with HEPES buffer (42 ± 2%), and substitution of 10 mM lactate in place of pyruvate did not reduce the 100 μM LA-induced LDH leakage compared with control HEPES buffer containing glucose alone (Figure 2C). The 2 mM concentration of CaE used is equivalent to 8 mg/dL calcium, and is thus clinically relevant [5]. Please note that buffers traditionally used for acinar studies [29], like ours, contain anions like pyruvate (10 mM). Retaining this concentration of pyruvate was however necessary to allow comparison with an equimolar concentration of sodium lactate (Figure 2C).

Cytosolic calcium (Cai) and mitochondrial depolarization (ψm) were then studied using Fura-2 AM- and JC-1-loaded acini and exposing them to 100 μM LA, as described previously [25,31], and comparing the effects of 0 mM (red), 1 mM (green), and 2 mM (purple) CaEs (Figure 2D,E). The increase in Cai over baseline by LA (added at 2 min) is shown. The LA-induced Cai at 1 mM was reduced significantly by 20–30% at 2 mM. The LA-induced ψm at 1 mM CaE was significantly lower than of 0 mM CaE, and was further reduced by 2 mM CaE between 400 and 600 s. This was consistent with chelation of CaE using ethylene glycol-bis (β-aminoethyl ether)-N,N,N′,N′-tetraacetic acid (EGTA; 1 mM) increasing LA-induced LDH leakage (Figure 2F). EGTA has previously been shown to increase LA-induced Cai [24]. These findings that CaE reduces lipotoxic injury were confirmed in the kidney cell line HEK293 (Figure 2G), which has been previously used to model renal injury in severe pancreatitis [36]. CaE dose-dependently neutralized the toxicity of LA (Figure 2G) in these, similar to pancreatic acini (Figure 2B). We next went on to study if CaE’s protective effect was unique to lipotoxic injury.

### 3.4. Physiologically Relevant Extracellular Calcium Concentrations Do Not Affect Caerulein-Induced Injury

We first compared the effects of increasing the CaE from 1 to 2 mM on cell injury induced by supraphysiologic (100 nM) caerulein in vitro. This simulates a milieu of pancreatitis for pancreatic acini. The 100 nM Caerulein-induced increase in Cai (Figure 3A) and LDH leakage (Figure 3C) was unaffected by increasing CaE from 1 to 2 mM. We could not detect any change in ψm over baseline induced by caerulein (Figure 3B). In contrast to CaE, chelation of intracellular calcium with 1,2-Bis(2-aminophenoxy)ethane-N,N,N′,N′-tetraacetic acid tetrakis-acetoxymethyl ester (BAPTA-AM; 50 μM) or antagonizing ryanodine receptors with dantrolene prevented caerulein-induced acinar injury (Figure 3D), as shown previously [37].

### 3.5. Antagonizing Intracellular Calcium Increase Has Little or No Effect on La-Induced Acinar Injury

Since antagonizing Cai increase prevented 100 nM caerulein-induced acinar injury, we went on to study whether this was relevant to LA-induced injury. For these studies, the acini were preincubated with BAPTA-AM (50 µM), Dantrolene (50 µM), or Thapsigargin (1 μM) for 10 min before the addition of LA. All three agents significantly reduced the Cai increase induced by LA (Figure 3E–G) by a large extent. BAPTA-AM had a small effect on the LA-induced ψm (Figure 3E’–G’), while the other agents had no effect. Interestingly none of the agents reduced LA-induced LDH leakage over 4 h significantly (Figure 3H). It is to be noted that the small protective effect of BAPTA-AM shown previously [24] did not reach statistical significance in the current study. Therefore, overall, unlike the 100 nM caerulein-induced injury mediated by the increase in Cai, that induced by LA in acini is largely independent of this increase of Cai. Overall, these studies show that CaE protects by by electrostatically interacting with LA, reducing its signaling in acini and consequent injury.

### 3.6. Therapeutic Extracellular Supplementation Binds Fatty Acids Generated During Severe AP

We next compared the in vivo efficacy of lactate and calcium during pancreatitis associated with organ failure, thus replicating severe pancreatitis in humans. The caerulein model in *ob/ob* mice (average 54.2 ± 1.5 gm), which causes organ failure in mice [23], was chosen. Caerulein (50 mcg/kg/h; CER) was delivered for 72 h as described in methods. We confirmed pancreatitis induction by verifying serum lipase (Figure 4A) to be 4–5 times above the control values 6 h after initiating pancreatitis. After this, we started Q8 hourly intraperitoneal administration of 1 mL of either 10 mM calcium chloride or 30 mM sodium lactate, both of which were in normal saline pH 7.4. The volume of these (55 mL/kg/24 h) is in the range used for IV fluid resuscitation in humans during severe pancreatitis [38,39]. This was designed as a survival study to retain its relevance to severe human pancreatitis, in which organ failure can occur late in the disease.

Administration of calcium, but not lactate, caused calcium deposition in the necrosed visceral fat of mice with pancreatitis given calcium (Figure 4B, CER + calcium, black arrows). Calcium soaps and noncovalent forms of calcium have been noted in pancreatic fat necrosis for over a hundred years [28,40]. On quantitative morphometry, calcium treatment reduced peri-fat acinar necrosis (PFAN), which is the necrosed acinar area adjacent to fat necrosis (3.0% ± 1.5% vs. 6.5% ± 2.2% in the lactate-treated group, *p* < 0.004; Figure 4C). Lactate itself did not influence the PFAN vs. caerulein alone (6.2% ± 0.9%), as previously shown [23]. Morphologically, calcium soaps were seen as von-Kossa-positive areas in the fat that stained intensely black for calcium in the animals given calcium (CER + Calcium, Figure 4D) during pancreatitis compared with those given lactate (CER + lactate, Figure 4D). Calcium or its salts stain black on von-Kossa staining. In adipose tissue, this signifies saponification, that is, covalent binding of calcium to the NEFA [41] generated by lipolysis of adipose triglyceride [24,27]. This saponification is not washed out by the organic solvents used in processing tissue for histology [42]. In mice administered calcium, this binding with calcium was seen as a clear black line adjacent to the fat necrosis (black arrows in CER + Calcium, in Figure 4D) which limited the PFAN. Such a demarcation was not seen in the lactate-treated group (Cer + lactate, Figure 4D von-Kossa), where we noted an extension of the PFAN deeper into the pancreatic parenchyma (black polygon in H&E of CER + Lactate, Figure 4D). Despite PFAN reduction in the calcium-treated group, there was no reduction in overall necrosis (13.0% ± 4.5% vs. 9.9% ± 2.7% *p* = 0.12), perhaps due to improved survival in the calcium-treated group (Figure 5A) causing a longer exposure to caerulein, as noted in the next section. This prolonged caerulein exposure caused acinar ductal metaplasia [43] in the calcium-treated group, seen as large luminal areas surrounded by flattened cells (asterisks in H&E of CER + calcium group, Figure 4D). This lack of reduced overall necrosis (despite the improvement in PFAN) in the calcium-treated group is consistent with CaE not affecting responses to supraphysiologic caerulein in vitro (Figure 3A,C). We next went on to analyze the effect of CaE on organ failure and pancreatitis severity parameters.

### 3.7. Therapeutic Extracellular Calcium Supplementation Prevents Hypocalcemia, and Transiently Reduces the Increase in Serum Non-Esterified Fatty Acids and Organ Failure

Caerulein AP resulted in a median survival of 48 h (black line, Figure 5A). This was significantly improved to 80 h in the calcium-treated group (green line), unlike the lactate-treated group (median survival 30 h, red line). Severe pancreatitis induced hypocalcemia similar to Ranson’s [5] and Glasgow criteria [6] (red line, Figure 5B). This was prevented in the group administered calcium. There was a large increase in serum nonesterified fatty acids (NEFA) on the second day of pancreatitis, which was significantly reduced in the calcium-treated group (shown as * in Figure 5C). This is consistent with CaE-binding NEFA generated in the peritoneal cavity during fat necrosis (Figure 4B,D). NEFA levels, however, subsequently increased over baseline (shown as # in Figure 5C), which is consistent with progression of fat necrosis over the 3 day course of pancreatitis, and the weak electrostatic interaction between calcium and NEFA, only a small part of which are covalently bound [28]. Similar to human studies [8,9], there was a significant increase in serum CRP noted on day 1, which was prevented by calcium (Figure 5D). Both renal injury (Figure 5E) and shock (Figure 5F) were significantly prevented till day 2 by calcium treatment. However, consistent with the later increase in NEFA, both of these worsened over the subsequent days. There was neither early nor late improvement in the serum IL-6 or TNF-α increase by any intervention (Figure 5G,H). Overall, these results show that calcium supplementation alone can replicate the early benefits noted with Ringer’s Lactate administration in clinical trials of AP.

### 3.8. Meta-Analysis of RCTs Comparing Normal Saline (Ns) Show a Reduction In Pancreatic Necrosis, but No Improvement in Long-Term Outcomes

The previous meta-analysis by Iqbal et al. [10] showed no improvement in mortality with RL vs. NS. Based on the three randomized controlled trials comparing the use of RL to NS in acute pancreatitis (Figure 6), we found that pancreatic necrosis was significantly reduced in the RL group with an Odd’s ration of 0.28 (CI = 95%, 0.09–0.92) (Figure 6B). ICU admission rate was not statistically different between the two groups. Three studies assessed the incidence of organ failure (OF). Choosakul et al. reported early OF within 48 h of treatment. A study by dDe Madaria et al. reported persistent OF, and Wu et al. reported organ failure without specification of timing. Overall, OF was not statistically different between the groups. Interestingly, the saline treatment arms had a higher number of patients (22 ± 2 vs. 20 ± 2, *p* = 0.0377) and a higher proportion of males (64 ± 10% vs. 46 ± 5%, *p* = 0.049) vs. the RL arms when accounted for in all trials (Figure 6A). This is important since some studies have shown males to be at a higher risk of severe acute pancreatitis [44,45,46,47], which may have contributed to the improved trends noted with RL. The meta-analysis’s findings of reduced pancreatic necrosis with RL correlate with the protection provided by extracellular calcium from fatty acid-induced acinar necrosis in vitro (Figure 2) and peri-fat acinar necrosis in vivo (Figure 4C,D). This pathophysiology is further supported by fat necrosis being a part of approximately 95% cases of pancreatic necrosis [48], and fat necrosis preceding pancreatic necrosis during human severe pancreatitis, based on observations [49] made during the era of early surgical intervention.

## 4. Discussion

Ringer’s lactate is widely used for resuscitative therapy during critical illnesses such as severe acute pancreatitis. Such illnesses are commonly associated with acute hypocalcemia [3,4,5,6]. In this study, we note that this hypocalcemia occurs due to NEFA like linoleic acid interacting with calcium (Figure 1). We note this as an ionic interaction, and previous studies have shown that a small part of this calcium is saponified [28,40], which we observe as von-Kossa-positive areas on histology (Figure 4). The extracellular calcium (CaE) provided by Ringer’s lactate thus reduces lipotoxic injury and improves early severity. These conclusions are based on mechanistic in vitro and in vivo studies comparing the effects of pH-adjusted solutions of lactate or calcium administered during severe pancreatitis. Calcium improved peri-fat acinar necrosis, BUN, CRP, and shock (Figure 5D–F) early on in the disease. This is similar to the improvements in SIRS, CRP, and pancreatic necrosis noted with Ringer’s lactate in two meta-analysis (ours and by Iqbal et al. [10]). Thus, the benefits noted with Ringer’s lactate [7,8,9,10] in patients with severe pancreatitis are likely due to the calcium it contains, which binds NEFA like LA with an ionic interaction and also covalently; the relative contributions of which remain unknown.

While we note improved organ failure and survival at the 48 h time point in the calcium-treated group (Figure 5A,E,F), the survival study design helped us realize that this was an early and transient improvement, as is noted in human severe AP treated with RL [7,8,9]. The early improvement in CRP is consistent with it being a protein released from the liver, and also by adipocytes [50,51], which generate NEFA during pancreatitis [24,27]. Such increases in CRP may be uniquely regulated in mice by IL-1 and not IL-6 [52]. Interestingly, lipotoxic severe pancreatitis is associated with increased IL-1β levels in both the necrotic collections and serum [24]. Thus, while the regulatory mechanisms affecting CRP levels are complex, this early reduction parallels the previous clinical studies [7,8,9,10], and the lower NEFA we note initially in the calcium-treated group. The early improvement in carotid pulse distention (Figure 5F) signifies improved blood pressure, and is also consistent with improvement of the SIRS, which was noted in patients given RL [7,8,9,10]. The reason underlying lack of improvement in IL-6 and TNF-α (Figure 5G,H) remains unclear, and could be due to macrophage activation being unaffected by calcium or lactate in vivo, despite lactate interfering with the TLR4-mediated effects [11], including preventing NF-κB activation [7], in these cell in vitro. Overall, it seems that since the main interaction of NEFA with calcium is ionic (Figure 1A–C), and covalent binding of calcium to NEFA is a small part [28]; the continuous NEFA generation during severe pancreatitis overrides the protective effects of calcium by day 3 (Figure 5), and the initial protection is not sustained.

The seminal study by Wu et al., which showed RL improves CRP and SIRS in AP, hypothesized avoidance of saline-induced hypercholoremic acidosis to be the main mechanism by which RL improves outcomes. This acidosis-based hypothesis was supported in their study by a smaller drop in serum bicarbonate in the RL group, and an inverse relationship between the serum bicarbonate levels and serum CRP. A higher bicarbonate in the RL group was also noted in the study by de-Madaria et al. [7], though there was no significant difference in the CRP or blood pH in subjects who received saline. In their study, saline and Ringer’s lactate caused a similar drop in pH of the culture medium to which these were added [7]. Based on these, we chose the vehicle for calcium or lactate to be saline with a final pH of 7.4, which is neither influenced by the ambient CO_2_, nor the respiratory or metabolic status of the organism. This vehicle bypassed the problem of a bicarbonate-based buffer being affected by respiratory, metabolic, or renal status during pancreatitis, and also that of a phosphate buffer reacting with and neutralizing the administered calcium before it reacted with NEFA. Our approach of using saline (pH 7.4) as a vehicle thus made the study agnostic to the pH in vivo, and is in agreement with a recent meta-analysis covering studies with >20,000 critically ill patients, which concluded that buffered solutions do not provide any survival benefit over normal saline [14].

The volume of saline or RL used in previous clinical trials when normalized to volume/kg/day has been 24–30 mL/kg/day to as high as 72 mL/Kg/day [7,8,9]. We chose a rate in between these values (55 mL/kg/day), since there is no consensus about the exact volume that should be used. This would be ≈ 3.8 L for a 70 Kg person, who would typically have a plasma volume of 2.8–3.2 L (https://www.mdcalc.com/blood-volume-calculation). Therefore, the amount of calcium in RL (Ca concentration 1.5 mM) could help in replacing calcium lost to binding with NEFA. For example, this volume could restore a Sr. Ca of 5 mg/dL (1.3 mM) to ≈10 mg/dL (2.5 mM) in a day in the absence of ongoing consumption by NEFA. The benefits of this are likely to be distinct from those of intravascular volume replacement, which is recommended in clinical guidelines. The intraperitoneal route we chose has pros and cons. The pros are (1) the agent is directly delivered to peritoneal fat necrosis to neutralize the fatty acids at source. This is based on human studies showing fat necrosis to be an early event in pancreatitis [49] and the mechanism of action we hypothesized in Figure 1. (2) This targeted delivery is not dependent on ischemia/perfusion state of the peritoneal fat, which could be influenced by the shock these mice develop (Figure 5F). (3) The delivery is effective in restoring serum calcium levels to normal values (Figure 5B). The cons are (1) we do not use this route in humans. (2) The calcium bound in saponification on the surface NEFA may block access of subsequent doses of calcium to the deeper layers of fat necrosis and NEFA formation. This may perhaps explain the later rise of serum NEFA (Day 3, 4 on Figure 5C), resulting in lack of sustained benefits from the administered calcium.

The designs of our in vitro and in vivo studies build on previous clinical studies showing unsaturated NEFAs, such as linoleic acid, to be increased in critical illnesses [15,16,17,18,19], the sera [20,21], and pancreatic collections [22,24,35] of patients with severe AP. In vitro, we note an energetically favorable interaction between linoleic acid and calcium but not lactate. The 1:8 stoichiometry of calcium and linoleic acid (Figure 1) suggests a weak noncovalent ionic interaction, which is supported by longer incubation periods with calcium progressively increasing protection of cells from lipotoxic injury (Figure 2A). This is also supported by an inverse relation between CaE concentrations and lipotoxic cell injury (Figure 2B,D,E,G) in acinar and HEK293 cells representing two organs injured in pancreatitis, that is, pancreas and kidney. Notably, calcium improved both BUN and peri-fat acinar necrosis during severe pancreatitis. Conversely, chelating CaE worsened this injury (Figure 2F), which is uninfluenced by the addition of lactate (Figure 2C). While we do not provide direct evidence of calcium supplementation increasing ionized calcium in the serum, it is logical that this contributed to the benefits of calcium supplementation, since the ionized pool of calcium reacts in the interaction we note, resulting in the beneficial effects of CaE.

The protection CaE provides has two clinically relevant unique aspects: (1) this protection is specifically from lipotoxicity, (2) the protection is not clearly linked to intracellular calcium (Cai) levels. The first point is highlighted by physiologically relevant CaE concentrations influencing lipotoxic- but not caerulein-induced injury (Figure 2 and Figure 3A,C). In vivo this translates to extracellular calcium not reducing acinar necrosis during caerulein pancreatitis despite reducing peri-fat acinar necrosis (Figure 4C,D). This is likely because the improved survival prolongs the caerulein exposure and resulting pancreatic necrosis, bringing the overall necrosis equivalent to the other groups, which have a shorter survival. In humans, this scenario may parallel an impacted gallstone causing persistent obstruction and cholangitis, resulting in sepsis from bacterial toxins, unresponsive to CaE or Ringer’s lactate. The second point is that CaE reacts with and reduces biological responses to linoleic acid (Figure 2D,E), including the increase in Cai (Figure 2D). However, inhibition of the linoleic acid-induced Cai increase by BAPTA, dantrolene, and thapsigargin (Figure 3E–G) does not translate to reduced cell injury (Figure 3H). This pattern is again unlike caerulein-induced injury, which is amelioratedby BAPTA-AM and dantrolene (Figure 3D), which reduce Cai [53]. Interestingly, all three agents that reduced linoleic acid-induced Cai had little or no effect on the ψm (Figure 3E’–G’), suggesting an independence of these two phenomena. This also supports the benefits of targeting CaE over Cai in reducing lipotoxic injury during human severe pancreatitis [21,27,54,55,56]. It is to be noted that calcium concentrations >1 mM do not inhibit lipolysis [57]. Thus, the protective effect of calcium is independent of it inhibiting lipolysis, and is mediated by its binding the NEFA generated from lipolysis.

In summary, we note that NEFA may cause acute hypocalcemia in critical illnesses, and extracellular calcium supplementation by Ringer’s lactate may contribute to the early clinical improvement in systemic inflammatory response and C-reactive protein noted during severe acute pancreatitis. This protection occurs via calcium reacting ionically with the fatty acids generated during pancreatitis, andtheir saponification. In future studies, it may be beneficial to compare the serum calcium levels in patients given calcium supplementation or Ringer’s lactate versus other solutions. Whether calcium supplementation or Ringer’s lactate can realistically provide any more than a short-lived early improvement in systemic inflammation and injury remains to be assessed in larger-scale human studies.

## Figures and Tables

**Figure 1 jcm-09-00263-f001:**
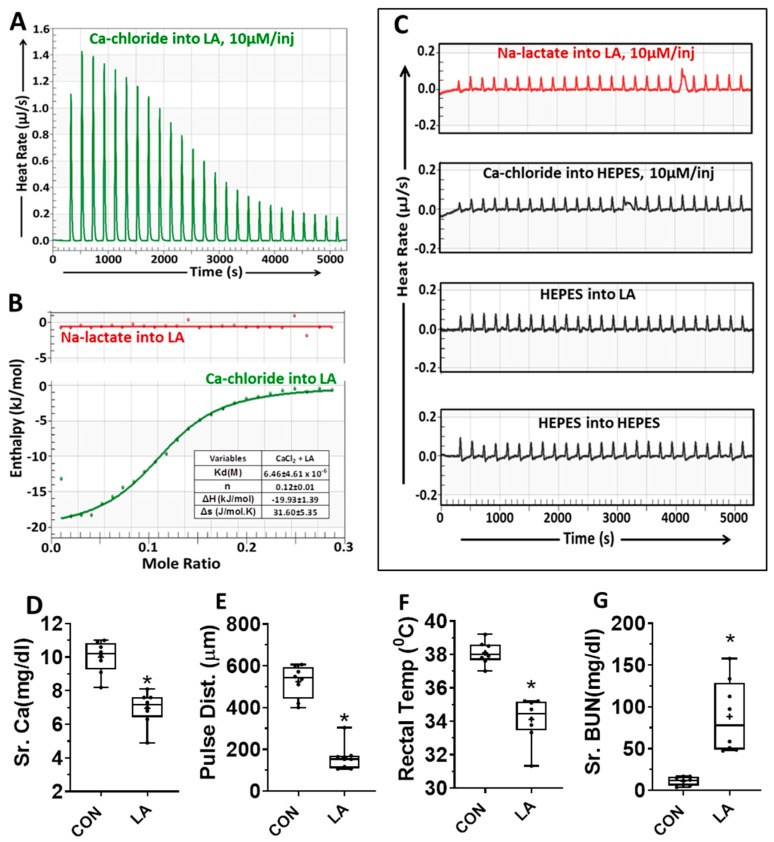
(**A**–**C**): Isothermal titration calorimetry showing Linoleic acid (LA) reacts with calcium but not lactate. (**A**) Thermogram showing heat rate of injection of calcium chloride (10 μM/injection) into 1 mM Linoleic acid (LA). Both were dissolved in 10 mM HEPES (pH 7.4) (**B**) Enthalpograms of injection of sodium lactate (Na-lactate, red line on top) or calcium chloride (CaCl_2_; green line below) into linoleic acid. The various thermodynamic variables (Mean ± SD (standard deviation)) from three different experiments are mentioned in the table adjacent to the thermogram. (**C**) Thermogram showing heat rate of injection of sodium lactate (top, red line; 10 μM/injection) and other control experimental conditions below this (black lines). (**D**–**G**): Effects of intraperitoneal administration of linoleic acid (LA, 0.1% body weight) to CD-1 mice on their serum calcium (**D**), carotid pulse distension (pulse dist.) (**E**), rectal temperature (**F**), and blood urea nitrogen (Sr. BUN) (**G**). The vitals were measured at 40 h, and blood parameters at necropsy.* indicates a *p*-value of <0.05 on Mann-Whitney test.

**Figure 2 jcm-09-00263-f002:**
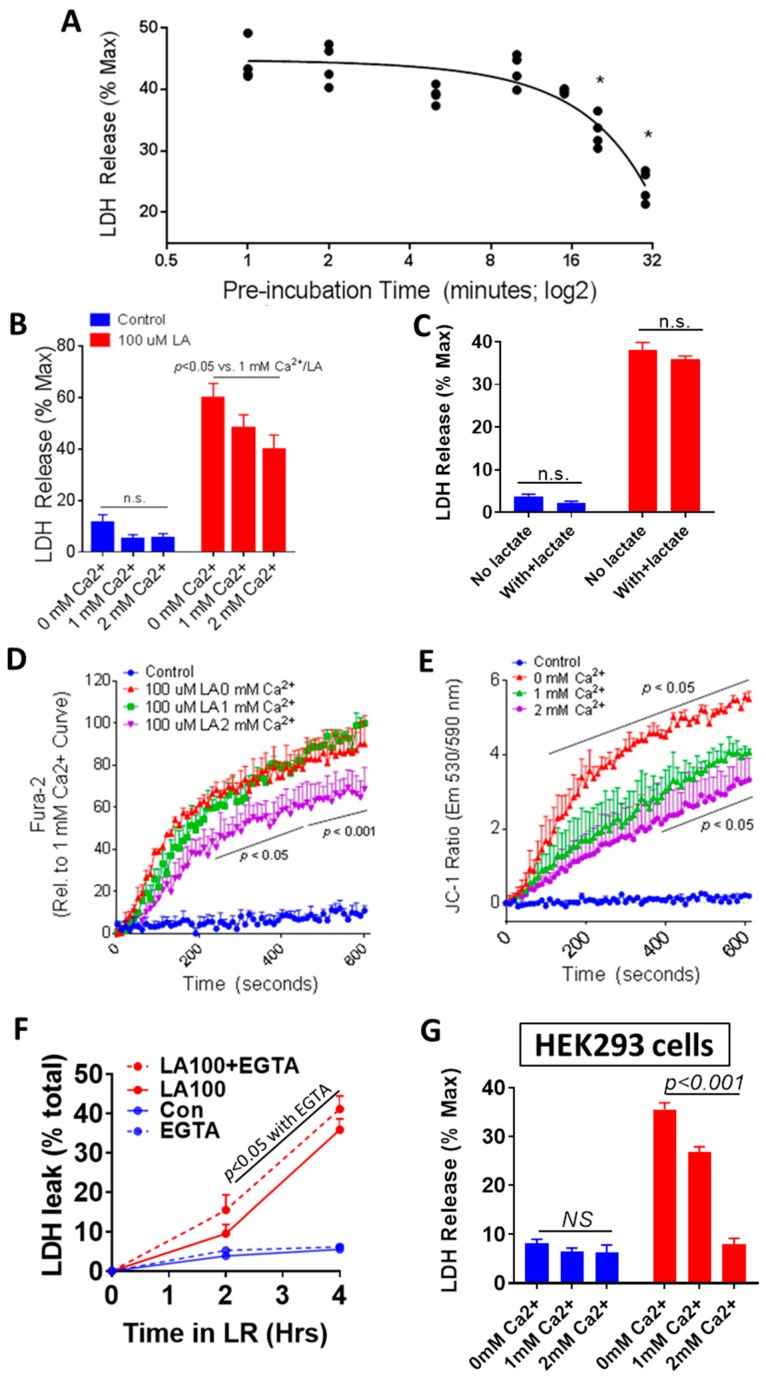
Extracellular calcium reduces LA-induced acinar cell injury in vitro. (**A**) Time course of the effect of different durations of preincubating LA with calcium at 37 °C prior to exposure to primary pancreatic acini, showing its impact on LDH release into the medium of the acini. (**B**) Extracellular calcium dose-dependently reduces 100 μM LA-induced acinar LDH release over 4 h (red bars) without affecting control cell viability (blue bars). (**C**) Supplementation with 10 mM sodium lactate had no effect on either baseline cell viability or 100 μM LA-induced acinar cell injury. Effect of different doses of extracellular calcium (0 mM red line, 1 mM green line, 2 mM purple line) on 100 μM LA-induced intracellular calcium elevation (**D**) and mitochondrial depolarization (**E**). (**F**) Chelation of extracellular calcium (1 mM) with EGTA (1 mM) to nominally absent levels increased LA-induced acinar cell injury (dotted red line) at 2 and 4 h. (**G**): Extracellular calcium dose-dependently reduces 100 μM LA-induced LDH release from HEK293 cells over 4 h (red bars) without affecting control cell viability (blue bars).

**Figure 3 jcm-09-00263-f003:**
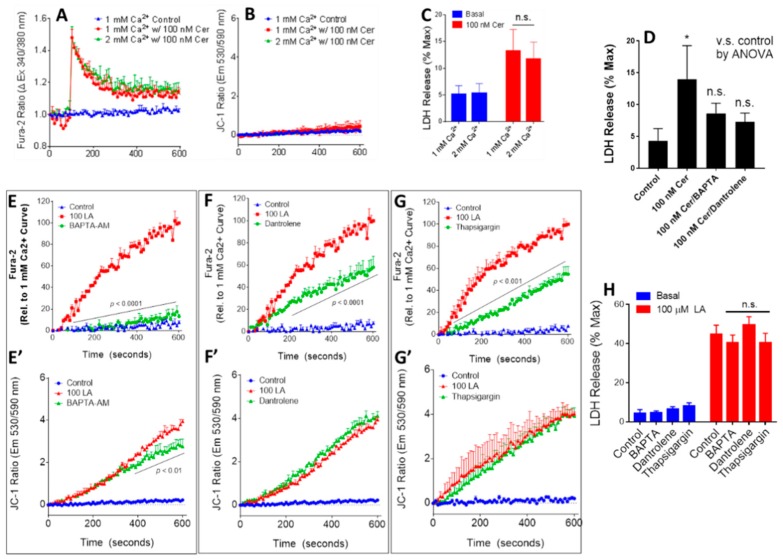
Effect of altering calcium on caerulein-induced (**A**–**D**), and changing Cai on LA-induced (**E**–**H**) acinar cell injury. (**A**) Extracellular calcium (1, 2 mM) had no effect on intracellular calcium peak and plateau levels induced by 100 nM caerulein and (**B**) did not modulate the lack of mitochondrial depolarization with caerulein treatment. (**C**) Extracellular calcium did not reduce caerulein-induced acinar cell injury, while (**D**) intracellular calcium chelation with BAPTA-AM (50 μM) or ryanodine receptor antagonization with dantrolene (100 μM) caused caerulein-induced injury to be nonsignificant versus control. (**E**) BAPTA-AM (50 μM), (**F**) Dantrolene (100 μM), and (**G**) thapsigargin (1 μM) all reduced intracellular calcium elevation after 100 μM LA treatment, while BAPTA-AM (**E’**) partially limiting mitochondrial depolarization Dantrolene (**F’**) and thapsigargin (**G’**) had no effect on mitochondrial depolarization. (**H**): Despite effects on calcium elevation, BAPTA-AM, Dantrolene, and Thapsigargin did not significantly affect LA-induced injury in acinar cells, as measured by LDH release.

**Figure 4 jcm-09-00263-f004:**
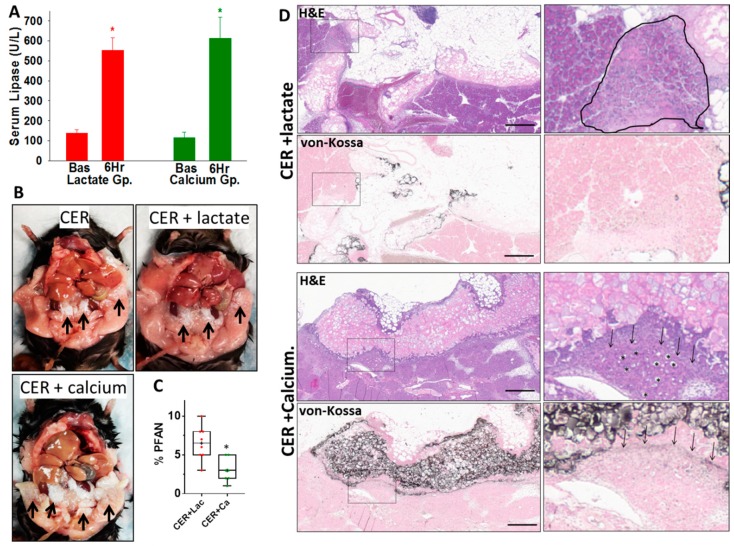
Comparison of the therapeutic calcium and lactate on fat necrosis during severe experimental pancreatitis: (**A**) serum lipase levels as measured under basal (Bas) conditions before induction of pancreatitis and after 6 h of pancreatitis (6 H), following which either Sodium lactate or Calcium chloride were given intraperitoneally, as described in the methods section. (**B**) Gross appearance of the peritoneal cavity at necropsy of mice with caerulein pancreatitis alone (CER)**,** caerulein pancreatitis treated with sodium lactate (CER + lactate), or caerulein pancreatitis treated with calcium chloride (CER + calcium). Note the much larger extent of white- colored saponification of fat necrosis in the calcium-treated group (black arrows). (**C**) Box plots comparing the extent of peri-fat acinar necrosis (% PFAN) measured as a percentage of total pancreatic parenchymal area in the CER + lactate (CER + Lac)- and CER + calclium (CER + Ca)-treated groups. * indicates a *p* < 0.05. (**D**) Histologic images of the pancreas and adjacent fat necrosis stained with hematoxylin and eosin (H&E), and von-Kossa for calcium, in the CER + lactate and CER + calcium groups. To the right are zoomed-in images of the inset boxes highlighting the peri-fat acinar necrosis adjacent to the fat necrosis. The scale bar is 200 μm. Fat necrosis resulting in binding of calcium to the NEFA generated is seen as the pinkish stain in adipose tissue on H&E. Note the higher intensity of this in the CER + calcium group using both types of staining, especially von-Kossa, which extends into the parenchyma as a sharply demarcated line (black arrows). The peri-fat acinar necrosis is seen as loss of acinar cell outline and diffuse pale pinkish appearance of acini replacing the normal intensely pink zymogen and blue-colored basal cytoplasm. In the CER + calcium group, which survive longer, the prolonged caerulein stimulation results in acinar ductal metaplasia-like appearance with large central lumens replacing the zymogen (shown as *).

**Figure 5 jcm-09-00263-f005:**
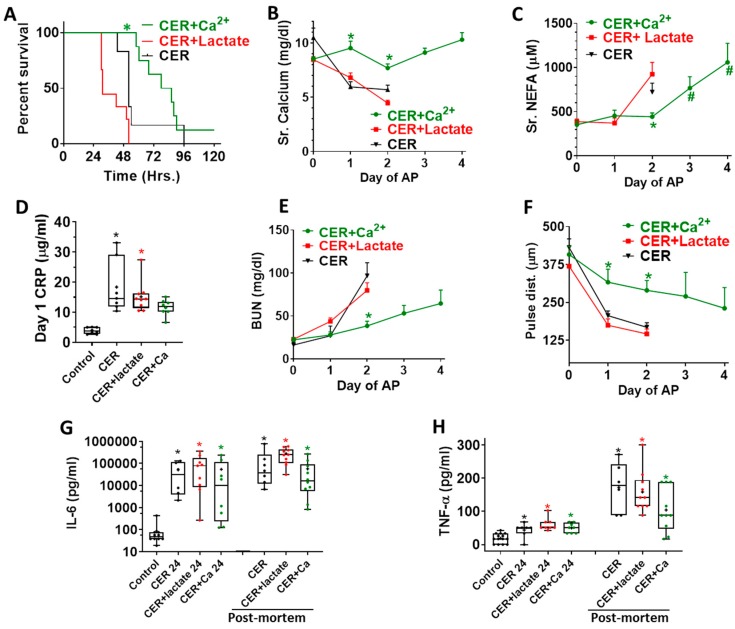
Effect of no treatment (black), calcium (green), or lactate treatment (red) on parameters of systemic injury and inflammation: (**A**) Kaplan–Meyer survival curve of the various groups with pancreatitis * indicates a *p* < 0.05 in the CER + calcium (CER + Ca^2+^)-treated group vs. other groups. (**B**) serum calcium and (**C**) Serum NEFA, as measured at each day of pancreatitis. * indicates a *p* < 0.05 in the CER + calcium treated groups vs. other groups on ANOVA. # indicates a significant difference from baseline (day 0). (**D**) Blood C-reactive protein (CRP) after 24 h of pancreatitis. * indicates a significant increase in the group on ANOVA vs. control mice without pancreatitis. (**E**) Blood urea nitrogen (BUN) and (**F**) Carotid artery pulse distention (pulse dist.), as measured on each day of pancreatitis. * indicates a *p* < 0.05 in the cer + calcium-treated groups vs. other groups on ANOVA. Interleukin 6 (IL-6) (**G**) and tumor necrosis factor (TNF-α) (**H**) blood levels, one day after pancreatitis initiation (each title followed by 24), and on the post-mortem blood samples. * indicates a significant increase on ANOVA for that time point vs. control.

**Figure 6 jcm-09-00263-f006:**
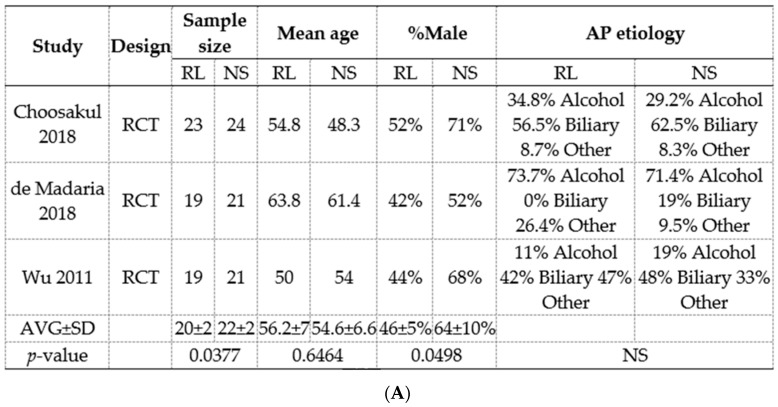
Meta-analysis of three randomized controlled trials comparing Ringer’s lactate to normal saline. (**A**) Study characteristics of those included in the meta-analysis. RL: Ringer’s lactate. NS: Normal Saline. AP: Acute pancreatitis. AVG: Average. SD: Standard deviation. NS: Not significant. (**B**) Outcomes of acute pancreatitis after hydration with Ringer’s lactate vs. normal saline. AP: Acute pancreatitis. RL: Ringer’s lactate NS: Normal Saline.

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
