# Peer review of "Ringer’s Lactate Prevents Early Organ Failure by Providing Extracellular Calcium"

_jcm, 2020, doi:10.3390/jcm9010263_

Round 1
Reviewer 1 Report
For several years since their seminal contribution in 2011 (Ref. 24), Singh and coworkers have published on the relevance of lipid breakdown (fatty necrosis, "lipotoxicity") in acute pancreatitis (AP). In the present study, they argue that Ringer-lactate solution is useful in AP due to its calcium content which binds free fatty acids (here: linoleic acid). This is an interesting piece of information, however, the study is presented in a confusing manner, and the results are mis- and over-interpreted in several instances.
Specific comments:
The abstract is confusing and should be completely rewritten. This is mainly an experimental study in acute pancreatitis, not a meta-analysis (which is added as Fig. 6). No methods are mentioned in the abstract, and the results are poorly summarized. The Methods part is catastrophic. Most important information is missing, including simple things e.g. how was the LA given to mice ? The mouse model (Fig. 4) is not described at all. There is no information on the fura-2 and JC-1 use, no information on any method used in vivo (e.g., carotid artery distension, NEFA and IL-6 measurements etc.). There are many discussions on pH in the manuscript, but most details remain obscure. Ringer-lactate solution (RL) usually has a pH of 6,5 – how can it correct acidosis ? What pH did the RL have that the authors used ? What is “saline pH 7,4” ? Sodium chloride in water does not have a pH of 7,4. Have the authors buffered the saline ? Ringer´s lactate contains 1,5 mM Calcium chloride which is lower than the calcium level in blood. How can it prevent hypocalcemia ? Figure 1 mixes data from a simple titration (A-C) with data measured in mice (D-G). This is confusing for the reader. What is more, the legends are missing. In Fig. 1A, a saturated fatty acid would have given the same result. In Fig. 1B, it does not make sense to “titrate” linoleic acid with lactic acid; instead, one could have measured the interaction between lactate and calcium. In Fig. 1D, it seems that the drop in calcium after injection of linoleic acid (LA) is not stoichiometric. Importantly, the data do not prove that a NEFA-calcium interaction was responsible for the observed pathology in mice; pancreatitis, shock and hypocalcemia occur in parallel, but that does not prove that they are causally related. This is a simple point of formal logic that should have been noted by the authors. In Fig. 2 pancreatic acini were damaged (LDH release) when exposed to LA. This toxicity is attenuated by calcium (1-2 mM), an effect that may be due to ionic interaction in this in vitro-experiment. There is also an increase of intracellular calcium and mitochondrial depolarization with LA which is again (moderately) attenuated by 2 mM calcium. It should be noted that 0 mM calcium is, of course, not physiological. The numbers mentioned in the text (e.g., lines 148-149) do not match the figures (e.g., Fig. 2C). EGTA – while not having much effect – increases LA toxicity (Fig. 2F). A similar effect could be seen in HEK cells (Fig. 2G). It should be noted that HEK cells are poor models of the kidney. In Fig. 3 the authors demonstrate that caerulein increases intracellular calcium but does not induce mitochondrial depolarization. The caerulein effect is sensitive to BAPTA and ryanodine. After stimulation with LA, the calcium modulators (BAPTA, ryanodine and dantrolene) reduced intracellular calcium but did not affect LDH release. The authors conclude that LA and caerulein have a different mode of action. Figure 4 is totally confusing, not least because legends and black arrows are missing and it remains guesswork for the reviewer which picture is which. Calcium does not affect pancreatitis (at least not lipase levels), but does it decrease or increase lipolysis ? The statement in lines 227-228 is that calcium increases “saponification” (a term that should be avoided and should be replaced by lipolysis). Does calcium increase fat breakdown but reduces necrosis ? Or is there an increase in necrosis ? Due to improved survival ? This speculation should not be made in the Results section. Generally, the Results part contains too much speculation - this should be reserved for the discussion. All in all, the pathological findings do not seem to match the in vitro-findings, and the various changes of necrosis, fat breakdown and survival seem complex and confusing. A clear relationship to the in vitro-findings is nowhere to be seen. Why were the mice not killed after a fixed time point for the pathology ? Or were they ? The text does not reveal this important fact. Figure 5 demonstrates an early beneficial effect of calcium (but not lactate) on caerulein-induced AP, and this is the best part of the study. Calcium-treated mice live longer (but die nevertheless), calcium levels are preserved and increases of fatty acids delayed, BUN remains lower and blood pressure higher for up to 4 days. However, the methods of measurement are not described, so it is impossible to understand the significance of the measurements. Moreover, I cannot follow the interpretation of these findings as a direct calcium-fatty acid interaction. In my view, calcium has some sort of stabilizing activity on pancreatic enzyme release and cell death, but its effect may, for instance, be due to lipid mediators formed from unsaturated fatty acids in the process of inflammation. A direct ionic interaction of LA with calcium is not demonstrated by these findings. Somewhat surprisingly, the levels of CRP, IL-6 and TNF-alpha were not changed by calcium administration (the statistics in Fig. 5D should be checked, in contrast to the authors I do not see a calcium effect). Finally, the authors present a meta-analysis of three clinical studies and find that RL administration is favorable to saline according to their definition of outcomes. This analysis does not contribute to the main focus of the paper because the RL used in these studies contained both calcium and lactate. This analysis may be helpful at the beginning of the manuscript to outline potential beneficial effects of RL. The discussion suffers from the author´s conviction that the fatty acid-calcium interaction must be responsible for beneficial effects of calcium. But the statement in line 329-330 is simply not true; the study does NOT show that the simple ionic interaction between calcium and a carboxylate group is responsible for complex in vivo-findings. Moreover, the authors confuse the reader with statements such as in lines 416-417 where the fatty acid-calcium interaction is suggested to cause “saponification”? Saponification is lipolysis, and free fatty acids cannot be “saponified” any further. Overall, the discussion is confusing and too wordy. It would be better if the authors just stick to their findings that the calcium in RL is responsible for an early protective effect in AP, including an effect on lipolysis / lipotoxicity. Discussions on the importance of pH are made difficult because the reasons given above. The further mechanistic discussion is also difficult because the authors demonstrated that fatty acids and caerulein act by different mechanisms, but then they use a caerulein model to investigate the effects of fatty acids in vivo. Typos: “fist” in line 111; “back” in line 237; “LR” in line 311.Author Response
Please see the attachment

Reviewer 2 Report
The manuscript by Khatua et al. describes the protective effect of calcium in Ringer’s lactate against pancreatitis severity. Using a variety of methods the authors demonstrate that calcium complexes with fatty acids and the resultant decrease in harmful fatty acid levels is protective in pancreatitis. The studies are well done and the documentation is convincing.
Major points.
The Methods section is very short; this should be either expanded or citations provided to methods used previously. There is some confusion with respect to the term “ionic interaction”. The reaction of calcium with fatty acids is only significant because the resultant salt is insoluble not because it is ionic. Not all anions will form insoluble salts with calcium (e.g. line 153, pyruvate??). Measurement of CRP as a marker of inflammation in mice is problematic as in this species CRP is not a strong acute phase protein. See Ku NO, Mortensen RF. The mouse C-reactive protein (CRP) gene is expressed in response to IL-1 but not IL-6. Cytokine 1993, 5:319-326.Minor typos:
Line 120, went on to study
Line 142, calcium
Line 172, minutes
Line 205, Since
Line 224, calcium chloride, sodium lactate
Line 312, odds ratio
Line 335, ours
Reviewer 3 Report
In this manuscript, authors provides evidence that Calcium rather than lactate present in Ringer’s lactate solution imparts the protection from early organ failure. Experimental evidence is based upon study of interaction of Ringer’s lactate components individually with free fatty acid linoleic acid, in vitro cell based and experimental model of severe acute pancreatitis.
Included experimental data amply support the conclusions of the study. Presence of calcium however, didn’t affect the production of inflammatory cytokines. Existing publications on acute pancreatitis related organ failure suggest that this injury might be triggered by multiple factors including free fatty acids. Given this, some of the protective effects of Ringer’s lactate might be calcium independent. Authors should expand discussion on this aspect.
The calculated stoichiometry of 1:8, for calcium and linoleic acid interaction is unusual, in view of proposed ionic interaction between the two. Possible explanation for this should be added. Authors should discuss if this interaction and calculated stoichiometry will hold, when in vivo linoleic acid will be present and interacting with other biomolecules.
Figure 1 & 4, labels for the panels (A, B, C……..) are missing.
Add method used to measure Carotid pulse distension in mice.
Round 2
Reviewer 1 Report
The authors have written several pages of comments to my review. In most cases, they defend their position and did not want to make major changes in style, presentation and/or discussion. They even cling to old-fashioned and chemically out-of-date nomenclature such as saponification. I do not think that further comments from my side would cause further changes.